# Beating Adolescent Self-Harm (BASH): a randomised controlled trial comparing usual care versus usual care plus a smartphone self-harm prevention app (BlueIce) in young adolescents aged 12–17 who self-harm: study protocol

Isobel Greenhalgh,[1] Jessica Tingley,[1] Gordon Taylor,[2] Antonieta Medina-Lara,[3] Shelley Rhodes,[4] P Stallard  [1,5]

¹Child and Adolescent Mental Health Services, NHS, Keynsham, UK
²College of Medicine and Health, University of Exeter, Exeter, UK
³Health Economics Group, University of Exeter Medical School, Exeter, UK
⁴Medical School, University of Exeter, Exeter, UK
⁵Health, University of Bath, Bath, UK

**Correspondence to**
Professor P Stallard;
p.stallard@bath.ac.uk

## ABSTRACT

**Introduction** A mobile app, BlueIce, was codesigned with young people with a history of self-harm to provide them with more accessible and available evidence-based support at times of distress. A preliminary evaluation found that BlueIce was acceptable, safe and used by young people and helped to reduce self-harm. The present study is designed to assess the effectiveness and cost-effectiveness of adding BlueIce to usual Child and Adolescent Mental Health Service (CAMHS).

**Methods and analysis** This study is a single-blind, randomised controlled trial comparing usual CAMHS care with usual care plus BlueIce. A total of 138 adolescents aged 12–17 with current or a history of self-harm will be recruited through the Oxford Health National Health Service (NHS) Foundation Trust via their CAMHS clinician. The primary outcome is self-harm at 12 weeks assessed using the Risk Taking and Self-Harm Inventory for Adolescents. Secondary outcomes include mood, anxiety, hopelessness, general behaviour, sleep and impact on everyday life at 12 weeks and 6 months. Health-related quality of life and healthcare resource utilisation data will be collected at baseline, 12 weeks and 6 months. Postuse interviews at 12 weeks will determine the acceptability, safety and usability of BlueIce.

**Ethics and dissemination** The study was approved by the NHS South Central—Oxford B NHS Research Ethics Committee (19/SC/0212) and by the Health Research Authority (HRA) and Health and Care Research Wales. Findings will be disseminated in peer review open-access journals and at academic conferences.

**Trial registration number** ISRCTN10541045.

## Strengths and limitations of this study

► This is the first randomised controlled trial to explore the effects of adding a self-help app (BlueIce) to usual care on the self-harm of adolescents receiving specialist mental health care.
► The study includes a detailed economic analysis to determine the cost-effectiveness of BlueIce.
► The BlueIce app was codesigned with young people with a lived experience of self-harm.
► Participants will not be blinded to participant group as this is prohibited by the nature of the intervention.

## BACKGROUND

Self-harm is defined as the deliberate act of causing damage to one's body, for example, through self-poisoning or self-cutting.[1 2] It is very prevalent, with around 17.2% of adolescents and 13.4% of young adults being estimated to self-harm.[3] Rates of self-harm are more likely to reduce in young adults compared with younger adolescents.[4] Most self-harm occurs in secret with comparatively few episodes, resulting in hospital presentations.[5–7] The most common reasons for self-harm include tension relief, escape from intolerable psychological pain, self-punishment and to show others how bad they are feeling.[7] While suicide is less prevalent among younger adolescents, it is the third most common cause of death in young people.[8] Self-harm has consistently been cited as a risk factor for later suicide attempts.[9] Studies have demonstrated that adolescents' non-suicidal self-harm is a strong predictor of future suicide attempts in young adults.[10] Findings such as these highlight the importance of intervening with those who are self-harming in preventing future suicides[11]

Research has identified risk factors for self-harm, including negative and stressful life events such as childhood maltreatment; self-harm or attempted suicide of a family member; drug and alcohol abuse; psychological factors such as feelings of hopelessness,

perfectionist traits, emotional dysregulation or diagnoses of psychological disorders; lower sociodemographic status and being female.[7 12–14] Protective factors which may help reduce or prevent self-harm include better access to social support, higher levels of self-esteem and receiving support from parents as well as being able to re-evaluate one's own thoughts and beliefs.[4]

Evidence-based interventions for self-harm in children and adolescents are scarce. Few studies have evaluated cognitive–behavioural therapy (CBT) or problemsolving.[15 16] Family therapy does not appear effective.[17] There is some support for dialectical behaviour therapy (DBT)[18–21] and for mentalisation-based treatment.[22] However, a Cochrane review concluded that there is not much evidence on which to draw conclusions on the effects of interventions for adolescent self-harm and recommended that therapeutic assessment, metallisation, dialectical behaviour and CBT warrant further evaluation.[15] Recent reviews have identified new studies evaluating DBT with adolescents and suggest that DBT does now meet criteria for a well-established treatment for self-harm.[23] With recent advances in technology, the use of digital interventions to aid the delivery of mental health interventions has become more widespread. The National Health Service (NHS) is encouraging the use of technology to improve access to, and the availability of, support and interventions, and to reduce demands on mental health services.[24]

There are currently over 15 000 mobile (m)health apps available worldwide, with around 5000 of these targeted at mental health.[25] Apps offer an accessible way of delivering and supporting mental health interventions for young people since over 80% of 12–15-year olds own a smartphone.[26] Alongside improving accessibility to support, mHealth apps are available 24/7, provide a means for symptom monitoring between face-to-face meetings, may be a preferable method of accessing support for some young people and provide a route around barriers such as stigma. mHealth apps may also help to lessen the demand on healthcare resources.[25 27 28] However, the evidence base surrounding healthcare apps is severely lacking, with research unable to keep pace with the speed of their development.[27] Therefore, the safety, efficacy and acceptability of most apps are unknown.

In terms of self-harm and suicide prevention, mhealth appears very acceptable and appealing to young people and offers a way of providing immediate support at times of crisis. However, hardly any suicide or self-harm prevention apps have been developed with a recent systematic review identifying only four.[29] Similarly, systematic reviews evaluating the efficacy of technology and mhealth interventions in preventing suicide or self-harm in adolescents[30] and university students[31] have raised similar concerns. In addition, the availability of self-harm prevention apps is extremely limited with few being available for clinical use.[31] While these reviews all highlight the potential of self-help apps, further research is required to determine their efficacy in self-harm and suicide prevention with clinical populations.

In response to the above, a self-help app, BlueIce, was codeveloped with young people with a lived experience of self-harm.[32] It is designed to support self-management of distress and reduce self-harm behaviours. BlueIce has 24/7 accessibility and is a prescribed app to be used alongside face-to-face Child and Adolescent Mental Health Services (CAMHS).[33] BlueIce is password protected and all data are stored locally on the phone.[32] It meets all minimum standards required for NHS accredited apps.[34]

An initial open study to determine safety, acceptability and usability found that 73% (19/26) of BlueIce users reported a reduction or cessation of self-harm at the 12-week follow-up.[27] No participants reported an increase in self-harm during the study. Postuse reductions in symptoms of anxiety and depression were also reported with 88% (29/33) of users electing to keep the app at the end of the study.[33]

The aims of this trial are to determine the effectiveness, cost-effectiveness and acceptability of adding BlueIce to usual face-to-face specialist mental healthcare in the reduction of self-harm in adolescents.

## METHODS

### Trial design

This is a two-arm, single-blind, randomised controlled trial (RCT) comparing the addition of the BlueIce self-help app to usual face-to-face specialist mental healthcare (usual specialist mental healthcare; UC +BI) with UC.

### Setting and participants

Participants will be recruited through CAMHS provided by Oxford Health NHS Foundation Trust, covering Bath and North-East Somerset, Swindon, Wiltshire, Buckinghamshire and Oxfordshire.

Young people will be eligible if they: (1) are receiving treatment from CAMHS at the time of referral, (2) have self-harmed at least two times in the last 12 months and (3) are aged between 12 and 17 years.

Exclusion criteria are: (1) a diagnosis of psychosis, (2) a significant learning disability which would interfere with the young person's ability to use the app, (3) young people with active suicidal plans or (4) safeguarding concerns where the young person has suffered abuse within the last 6 months or is the subject of a safeguarding investigation.

### Recruitment

Clinicians across Oxford Health NHS Foundation Trust will be invited to identify eligible young people who are open to CAMHS. Clinicians will discuss the study and provide interested young people, and if under 16 their parents/carers, with a project information sheet. If interested in participating, their details will be forwarded to the research team.

### Consent

Researchers will meet with the young people, and if appropriate their parents/carers, to discuss the project. If the

young person wants to take part, the researcher will then obtain consent. If under 16 years old, the young person will be asked to provide assent while their parent or legal guardian will be asked for consent. Those aged 16 years or older will be able to provide their own signed consent. During the COVID-19 pandemic, the consent and assessment process will be undertaken remotely (ie, online or via telephone) to maintain the safety of participants and the research team.

### Randomisation

Computer-generated randomisation will be independently undertaken by Exeter Clinical Trials Unit. Participants will be randomised in a 1:1 ratio to either UC or usual UC +BI. Participants will be randomised using REDCap software minimising for gender, age (over or under 16), self-harm frequency in last 4 weeks (0–2 or ≥3) and severity of depression (Mood and Feelings Questionnaire (MFQ) above or below 27, the cut-off for severe depression). Because of the nature of the intervention, participants will not be blind to their allocation. However, researchers involved in data collection will remain blind to allocation. Participant allocation will be undertaken by a member of the project team who is not involved in undertaking patient assessments.

### Interventions

UC: Young people will receive individual mental health interventions from specialist CAMHS clinicians. This will be either face-to-face, or, due to COVID-19, a remotely delivered telephone or video intervention. The nature, content and duration of this will be captured by the healthcare resource questionnaire.

UC+BI: in addition to usual care, young people will also receive access to the self-help BlueIce app.

BlueIce is an application for android and apple smartphones. It contains a mood diary, personalised toolbox of mood lifting strategies that are available to the young person 24/7 and automatic routing to emergency contact numbers.

### Mood diary

On entering BlueIce, young people rate their mood. For each mood rating, the young person has the option of adding a note to record any particular reason why they might be feeling as they do. Their rating and notes are saved in a calendar, which the young person and therapist can review to look for changes and patterns over time.

### Mood lifting

If the young person rates their mood as low, they will automatically be routed to the mood lifting section. Alternatively, if at any time the young person would like to access this section, they can do so directly from the main menu. This section contains a menu of mood lifting and distress tolerance activities, personalised according to the interests of the young person. The activities are designed to counter the common reasons why young people self-harm (to punish themselves, emotional relief, feeling hopeless) and draw on common methods used in CBT and DBT. The mood-lifting section includes eight activities: (1) photo library: the young person can upload and save photographs, inspirational quotes and pictures that are associated with happy memories or which might make them feel good. These can be reviewed when low to help the young person remember the positive things in their life; (2) music library: a music player is included where the young person can upload and store music they enjoy, and which has a positive effect on how they feel. This playlist can be readily accessed when the young person is low as a way of improving their mood; (3) physical activities: the young person can identify physical activities they enjoy such as sporting activities (eg, going for a run or riding a bike) or other aerobic activities such as walking the dog. The young person can access their personalised list when low and be reminded about what they can do to get active to improve their mood; (4) mood changing activities: BlueIce includes a section of activities that make the young person feel good. These could be things like making a cake, watching an episode of a favourite TV series, reading a book, playing with a pet. These provide the young person with a prompt list of activities they can use to change their mood when feeling down; (5) relaxation and mindfulness exercises: audio-recorded instructions for a 10 min mindfulness session, calming visualisation and a quick controlled breathing exercise (4–7–8 breathing) are included. These can be used to help the young person manage any unpleasant emotions or distressing thoughts; (6) identification of negative thoughts: this section includes a thought diary where the young person can record any troubling thoughts that are racing through their head. These can be directly typed into BlueIce where they are saved and can be reviewed at a later date. This allows identification of any themes that could be addressed during face-to-face work with their clinician; (7) ride it out: this section draws on ideas from DBT and helps the young person to tolerate their distress. This includes instructions for an ice dive, a sensory toolbox and a 'pros and cons' balance sheet for self-harming; (8) call a friend: the final section contains the phone numbers of 3–5 people who the young person could contact if they were feeling low and in danger of self-harming. These would be people who make them feel happy and those they could talk with about how they are feeling. This section prompts the young person to reach out to others.

### Emergency contacts

After accessing the mood lifter, the young person is asked to rerate their mood. If they are still low and feeling that they might harm themselves, they will be routed through a series of questions to three emergency contact numbers. The young person can select one of these options to automatically call/text emergency support.

## Patient and public involvement

BlueIce was codesigned and produced with young people with a lived experience of self-harm. They were involved in exploring the concept (would an app be helpful?), what an app should look like (examples of apps liked and used), the design (font, colours, flow) and content (evidence based and ideas young people found helpful).

In this study, two young people will be recruited to join our Study Steering Committee (SSC). We plan participant workshops to develop study resources, to advise on recruitment and retention issues, to discuss study findings, identify key messages, prepare understandable research summaries in different formats and identify issues/implications for future research. We intend to involve young people in events disseminating the findings of the study.

## Assessment schedule

Data will be collected at: (1) baseline; (2) postintervention (12 weeks) and (3) follow-up (6 months after randomisation). Data will be collected by research assistants, blind to treatment allocation. Participants will be given a £20 voucher after completing the final assessment.

### Baseline

Standardised self-report measures of self-harm, depression, anxiety, hopelessness, general behaviour, impact on everyday life, sleep and health-related quality of life and resource use healthcare questionnaire. These will be complemented with case-note review: to detail resource use, that is, accident and emergency department attendances, out of hours contacts, primary and secondary care attendances following incidents of self-harm in the preceding 6 months.

### Postintervention (12 weeks)

Standardised self-report measures of self-harm, depression, anxiety, hopelessness, general behaviour, impact on everyday life, sleep and health-related quality of life and the resource use healthcare questionnaire will be repeated (baseline—12 weeks). Those in UC +BI will complete a semistructured interview detailing their use, experience of, and satisfaction with BlueIce.

### Follow-up (6 months after randomisation)

Standardised self-report measures of self-harm, depression, anxiety, hopelessness, general behaviour, impact on everyday life, sleep and health-related quality of life. Case note review will be repeated (12 weeks to 6 months), and the type and total hours of direct and indirect CAMHS intervention provided from randomisation to 6 months are detailed.

## Outcome measures
### Primary outcome

Our assessment of self-harm will consist of three parts: (A) a brief interview, (B) completion of the Risk Taking and Self-Harm Inventory for Adolescents (RTSHIA)[35] and (C) the provision of support and advice.

### Part A: interview

Young people will be asked 'have you ever hurt yourself on purpose in any way (eg, by taking an overdose of pills or by cutting yourself) over the past 3 months?' which was taken from the Avon Longitudinal Study of Parents and Children (http://www.alspac.bris.ac.uk).[36] Young people are asked to consider the last 6 months at baseline and the last 3 months at follow-ups. Those who answer yes will be asked further questions about frequency, method, reason for self-harming, whether they sought medical help and suicidal intent.

### Part B: RTSHIA

Our primary outcome is self-reported self-harm assessed by the self-harm inventory of the RTSHIA. The RTSHIA was developed in the UK for use with adolescents (aged 11–19 years).[35] The self-harm inventory consists of 18 items and assesses the presence and frequency of a range of intentional self-injuries (eg, cutting, burning, self-hitting, self-poisoning). The frequency of each item is rated on a 4-point scale (never, once, more than once and many times) over a defined period. At baseline, the young person is asked to consider the 6 months before their initial assessment. For the 12-week and 6-month assessment, the young person considers the 3-month period since their last assessment. Each item is then scored (0, 1, 2, 3) and totalled to provide a current self-harm score. The RTSHIA has good reliability and validity.[22 35] We will analyse total scores and use this information to categorise changes in self-harm from baseline to 12 weeks and 6 months as reduced/stopped versus same/increased.

### Part C: support and advice

At the end of the assessment, young people will be given a list of contacts they can call if they are feeling worried about themselves. These include NHS 111, Childline and the Samaritans.

### Secondary outcomes

The MFQ[37] is a self-report questionnaire for depression recommended by National Institute for Health and Care Excellence (NICE) consisting of 33 items rated as either 'true' (scores 2), 'sometimes true' (scores 1) or not true (scores 0). The MFQ has high criterion validity and correlates well with other measures of depression.[37] A total score of 27 and above is associated with major depression, 20 with mild depression and 16 with no mood disorder. The Hopelessness Scale for children, adapted from Beck's Hopelessness Scale,[38 39] consists of 17 true–false items measuring hopelessness and negative expectations for the future. Items endorsed as 'true' are summed, with higher scores indicating greater hopelessness. The Hopelessness Scale for children has been widely used within adolescent samples and has consistently demonstrated strong psychometric properties.[38 39] The Revised Child Anxiety and Depression Scale[40] is a 47-item questionnaire with items corresponding to DSM-IV criteria for anxiety in the areas of social phobia, separation

anxiety, obsessive compulsive disorder, panic disorder, generalised anxiety disorder and for major depressive disorder. Each item is rated on a 4-point Likert scale of frequency (never 0; sometimes 1; often 2; always 3), which are summed to produce subscale and total anxiety scores. The Strengths and Difficulties Questionnaire[41] is a widely used behavioural screening questionnaire consisting of 25 items assessing emotional symptoms, conduct problems, hyperactivity and/or inattention, peer relationship problems and prosocial behaviour. Each item is rated as not true (0), somewhat true (1) or certainly true (2). A total difficulty score is calculated by summing scores from all subscales except the prosocial. In addition, an impact supplement assesses the degree of distress created by the child's difficulties and the degree to which they interfere with home life, friendships, classroom learning and leisure activities. The five items are summed to produce a total 'impact on everyday life' score, which ranges from 0 to 10. The Sleep Condition Indictor (SCI)[42] is an eight item self-report measure, assessing sleep and impact on daytime functioning over the past month on a 5-point scale. Item scores are summed to produce a total score ranging from 0 to 32. The SCI is an internally consistent ($\alpha = 0.86$) measure with a clinical cut-off <17 correctly identifying 89% of those with probable DSM-5 insomnia disorder.[42]

## Qualitative evaluation

We will use the semistructured interview developed in our initial study[43] to assess participant's experience of BlueIce, including use, ratings of satisfaction, helpfulness, ease of use and whether they would recommend it to a friend. In addition, we will assess the degree to which BlueIce was used and which parts of the app were used most often. This will only be completed by UC+BI at 12 weeks.

## Economic analysis

A cost-effectiveness analysis will be undertaken alongside the RCT to estimate the incremental cost-effectiveness of UC+BI compared with UC, that is, incremental cost per unit of health outcome (primary outcome) and quality-adjusted life years (QALYs) using the NHS and social care perspective. In order to estimate QALYs, we will use the Child Health Utility 9D (CHU-9D). This preference-based generic HRQoL measure is designed specifically for use in the economic evaluation of healthcare interventions in young people.[44] The CHU-9D contains nine dimensions (worried, sad, pain, tired, annoyed, school-work/homework, sleep, daily routine and activities), each with five levels of functioning rated for 'today'. The CHU-9D has been validated for self-completion by young people (aged 7–17 years)[45] and with CAMHS.[46] Resource use and cost data will be collected from the participants recruited in the RCT using a resource use questionnaire. Data on self-reported resource use will be compared with the self-harm assessment data that are recorded in clinical case notes (CareNotes). Clinical records will be reviewed for accident and emergency attendances, out of hours

contacts or primary and secondary care attendances following self-harm over two periods (6 months to baseline and baseline to 6 months). We will also quantify the number of face-to-face appointments and total number of hours of CAMHS input provided from baseline to 6 months. Incremental costs will be combined with data on effectiveness/health outcomes. National unit costs will be obtained from available sources including Personal Social Services Research Unit (PSSRU) (https://www.pssru.ac.uk/) and from the National NHS cost collection (https://www.england.nhs.uk/national-cost-collection/). Analyses will follow good practice for conducting economic evaluations in health technology assessment[47] and findings will be reported using the CHEERS guidelines.[48] Results will include disaggregated data, as well as synthesis of cost and outcome data, and will include presentation of cost-effectiveness plane,[49] cost-effectiveness acceptability curves[50] and detailed consideration of the broader impacts of the results reported. Robustness will be assessed through sensitivity analyses. Multiple imputations will be used to 'fill-in' missing cost and outcome data, making the assumption that the data are missing at random.[51] If the young person is under 16, we will also ask their parent or guardian to complete both these questionnaires.

## Sample size

A 3-point difference on our primary outcome (RTSHIA) between treatment groups represents a clinically important difference.[35] However, we propose to adopt a more conservative approach and will power the study to detect a moderate effect representing a 2-point difference. With an SD of 3.6, 90% power, alpha set at 0.05, we will require 69 participants per group.

## Planned analysis
### Statistical analysis

Our primary analysis will be at the end of the 12 weeks follow-up of the last recruited participant. A statistical analysis plan will be developed by the trial statistician in consultation with the project management group and agreed with the SSC before database lock. We will follow the Consolidated Standards of Reporting Trials extension for reporting RCTs and will follow recommended guidelines for analysis of our data.[52] Our primary analysis at 12 weeks will be analysed on an intention to treat principle. Although we are not expecting a significant amount of missing data at 12 weeks, the impact of missing data will be assessed by comparing baseline covariates for missing and non-missing cases. In the event that there is evidence of bias being introduced into the analysis, then further consideration, including but not limited to multiple imputation, will be given regarding how to address this.

Descriptive statistics will summarise baseline characteristics for each arm and patterns of missing follow-up data will be explored. We will also undertake a per protocol analysis of our primary outcome, total scores on the RTSHIA. Regression analysis adjusting for baseline minimisation variables of age, gender, mood and self-harm

frequency will be undertaken. We will conduct sensitivity analyses, in which we adjust for prognostic variables for which there is a baseline imbalance between intervention arms. Further sensitivity analyses will use multiple imputation to deal with missing data.

Similar regression analyses will be conducted for secondary outcomes (linear regression for numerical outcomes and logistic regression for binary outcomes). All secondary outcome measures will be compared between the groups and will include summary statistics and CIs for measures of effect size.

Analysis of the 6-month data will be included in a repeated measure analysis to investigate the maintenance of any effect seen at 12 weeks. Analysis of the 6 month follow-up data will be undertaken using a repeated measures analysis of variance with both the 12 weeks and 6-month data being included and adjusted for the baseline. The analysis will also be adjusted for the baseline minimisation variables: age, gender, mood and frequency of self-harm as proposed for the primary analysis at 12 weeks.

### Trial management

An independent Study Steering Group (SSG) will be established to monitor progress, advise the investigators in general scientific and management issues and ensure that there are no major deviations from the study protocol. The SSG will include an independent chair, and at least two other independent members with research experience with young people with mental health problems and/or self-harm. The SSG will also include two young people from the Oxford Health participation group. The SSG will meet at least once per year. The lead applicant will inform the SSG Chair who may call additional meetings when there are matters arising from the conduct or management of the trial that might require their advice

A Data Monitoring Committee was not convened as there was no panned interim analysis. Adverse events were reviewed by the SSG.

### Adverse event reporting and harms

For the purposes of this trial, adverse events are defined as increases in extent of self-harm or suicidal ideation regardless of whether they are casually related to the trial procedures. Serious adverse events are those requiring hospital admission. All adverse and serious adverse events will be reported to the project leader and will be reviewed by the SSC.

Clinicians will be requested to inform study researchers should a young person experience an adverse event during their participation in the study. At both follow-up visits, the young person (and parent if under 16) will be asked whether any adverse events have occurred and whether they think the study is having any negative effects on their mental health and self-harm frequency.

### Ethics and dissemination

Favourable ethical opinion for the research was obtained from the South Central—Oxford B NHS Research Ethics Committee (19/SC/0212) and was approved by the HRA and Health and Care Research Wales, prior to the recruitment of participants commencing at any NHS site. We will disseminate our findings to academics and researchers through high impact open access publications and through presentations at relevant academic and clinical conferences. Results will be made available to all participants after the completion of the study.

## DISCUSSION

There is little research evaluating the effectiveness and cost-effectiveness of interventions in the treatment of self-harm in adolescents. Intervening with those who self-harm is important since self-harm during adolescence is a significant risk factor for future suicide attempts and completed episodes.[10 11] The development of mHealth apps to support mental health interventions for adolescents offers a novel and accessible way of providing support at times of crisis. However, while technology offers many potential benefits, few apps to prevent self-arm have been developed and there is a lack of evidence regarding their efficacy, safety and acceptability.[25]

### Strengths and limitations

This is the first adequately powered RCT of a digital self-help app codesigned with, and used by, young people aged 12–17 years in receipt of specialist mental health services. This study will add to the evidence-base and will document the effectiveness and cost-effectiveness of adding a self-help app to usual care. If benefits are identified, BlueIce app can be widely made available to young people attending specialist CAMHS to provide help at times of distress.

As it is not feasible to blind participants to allocation, the research team is aware of the need to maintain equipoise and to present the two interventions to referring clinicians and participants in a balanced way. Researchers will remain blind to treatment arm and, if this is inadvertently broken, subsequent assessments will be conducted by another member of the research team. Similarly, following guidance from our participation group, those allocated to usual care will be provided with BlueIce at their final, 6-month assessment.

We have minimised the possibility of contamination between the trial arms. BlueIce is a prescribed app and is not freely available to download and use. Those allocated to UC+BI will be sent a single use download code. Once activated, BlueIce will automatically be installed on the participant's smartphone and the access code will no longer work. Participants are, therefore, unable to share/pass the app/access code to others.

Finally, this project will evaluate the use of BlueIce as a prescribed app, used in conjunction with a specialist mental health intervention. Future research will be

required to evaluate the use of BlueIce as a standalone, freely accessed self-help app or the mechanisms underpinning any effects.

## Trial status

Recruitment began in January 2020 and, pending COVID-19, will end around June 2022.

## Trial sponsor

Oxford Health NHS Foundation Trust. The sponsor will have no role in interpreting data, writing reports or decisions to publish findings.

**Contributors** PS, GT, AM-L and SR designed and obtained funding for the study. PS, IG and JT drafted the study protocol. All authors contributed and approved the final version.

**Funding** This project is funded by the National Institute for Health Research (NIHR) under its Research for Patient Benefit (RfPB) Programme (grant reference number NIHR/PB-PG- 1217–20004).

**Disclaimer** The views expressed are those of the authors and not necessarily those of the NIHR or the Department of Health and Social Care. The funder will have no role in interpreting data, writing reports or decisions to publish findings.

**Competing interests** BlueIce is the intellectual property of Paul Stallard, the creator of the app. He has no financial benefits from the app.

**Patient and public involvement** Patients and/or the public were involved in the design, or conduct, or reporting, or dissemination plans of this research. Refer to the Methods section for further details.

**Patient consent for publication** Not applicable.

**Provenance and peer review** Not commissioned; externally peer reviewed.

**Open access** This is an open access article distributed in accordance with the Creative Commons Attribution 4.0 Unported (CC BY 4.0) license, which permits others to copy, redistribute, remix, transform and build upon this work for any purpose, provided the original work is properly cited, a link to the licence is given, and indication of whether changes were made. See: https://creativecommons.org/licenses/by/4.0/.

**ORCID iD**
P Stallard http://orcid.org/0000-0001-8046-0784

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
