## [Reviewer comments · BMJ Open]

ARTICLE DETAILS

TITLE (PROVISIONAL)	Beating Adolescent Self harm (BASH). A randomised controlled trial comparing usual care versus usual care plus a smartphone self-harm prevention app (BlueIce) in young adolescents aged 12-17 who self-harm: study protocol.
AUTHORS	Greenhalgh, Isobel; Tingley, Jessica; Taylor, Gordon; Medina-Lara, Antonieta; Rhodes, Shelley; Stallard, P

VERSION 1 – REVIEW

REVIEWER	Clarke, Stephanie Stanford University
REVIEW RETURNED	05-Apr-2021

GENERAL COMMENTS	- DBT is a well-established, evidence-based intervention for self-harming adolescents at high risk for suicide. It is the only treatment meeting this threshold and should be discussed as such so as to not mislead readers.- discuss why youth self-harm in intro- individuals can experience self-harm urges in the absence of low mood - were participants only given replacement skills for self-harm if mood was rated as very low?- note that DBT distress tolerance skills (e.g., ice dive) are not meant to enhance mood. They are aptly included under "ride it out" section here, but should not be lumped under a larger heading of "mood lifting activities."- if participants only have to have two episodes of self-harm, I'm not sure that decrease or elimination of self-harm will answer your question of whether the app helps to reduce self-harm episodes.
--

REVIEWER	Forte, Alberto Sapienza University of Rome
REVIEW RETURNED	06-Apr-2021

GENERAL COMMENTS	I am thankful for the opportunity to review the present study protocol. I find that the present study is very important and up to date, as there is a lack of evidence-based mobile apps to be used with the aim of preventing self-harm in adolescents. Moreover, the previous study already demonstrated the feasibility and acceptability of the BlueIce app. The methodology is very clearly described, the paper is well written and readable. I would only recommend the authors providing some more details on the importance of the present study. I would highlight the utility of mobile devices for suicide prevention in adolescents, and also the lack of studies on the effectiveness. A recent review paper highlighted this aspect and might be useful for the introduction
---

	section (Medicina (Kaunas). 2021 Jan 26;57(2):109. doi: 10.3390/medicina57020109.). I would also suggest moving the description of Blulce in the method section rather than the introduction, as I would use the introduction to specify the importance of the present study given the lack of such investigations. Also, the introduction is weak in providing information and importance of the problem; self-harm is not only a risk factor for suicide attempts, but those who self-harm are at risk for later completed suicide (more recent citation: Hawton, K., Bergen, H., Cooper, J., Turnbull, P., Waters, K., Ness, J., & Kapur, N. (2015). Suicide following self-harm: Findings from the Multicentre Study of self-harm in England, 2000-2012. Journal of Affective Disorders, 175, 147–151. https://doi.org/10.1016/j.jad.2014.12.062). Nonetheless, these aspects might be further included in the discussion section, which seems to be too synthetic.
--	---

VERSION 1 – AUTHOR RESPONSE

Reviewer: 1 Dr. Stephanie Clarke, Stanford University Comments to the Author:

- DBT is a well-established, evidence-based intervention for self-harming adolescents at high risk for suicide. It is the only treatment meeting this threshold and should be discussed as such so as to not mislead readers.

- We have not attempted to mislead readers about the evidence base for DBT. In the UK we are informed by the National Institute for health and Care Excellence (NICE) and DBT is NOT yet a recommended treatment for adolescents. Similarly, the more recent Cochrane reviews we cite state that there is little evidence to draw conclusions on the effects of any interventions. What we have stated is therefore accurate.

We note the reviewers 2019 paper and the more recent research and reviews they cite. We acknowledge that the evidence base for DBT is growing and have therefore cited the reviewer's paper and have added the following: "Recent reviews have identified new studies evaluating DBT with adolescents and suggest that DBT does now meet criteria for a well-established treatment for self-harm".

- discuss why youth self-harm in intro

- We have added the following sentence in the introduction. "The most common reasons for self-harm include tension relief, escape from intolerable psychological pain, self-punishment and to show others how bad they are feeling".

- individuals can experience self-harm urges in the absence of low mood - were participants only given replacement skills for self-harm if mood was rated as very low?

- No. Young people who enter the study will be self-harming and we anticipate that most will also have significant symptoms of depression. In our feasibility study, 96% of participants also had clinically significant symptoms of low mood suggesting depression.

- note that DBT distress tolerance skills (e.g., ice dive) are not meant to enhance mood. They are aptly included under "ride it out" section here, but should not be lumped under a larger heading of "mood lifting activities."

- We agree that DBT strategies are more about emotional tolerance rather than mood change. However, the app was designed with one general "emotional management/coping section" rather than separate mood changing and distress tolerance sections. It was based on advice from our young

people's participation group that this section, including DBT skills, was called mood lifting. We have clarified this in our description of Bluelce: "This section contains a menu of mood lifting and distress tolerance activities"

- if participants only have to have two episodes of self-harm, I'm not sure that decrease or elimination of self-harm will answer your question of whether the app helps to reduce self-harm episodes.

- Thank you for your comment. This is one of the eligibility criteria for inclusion in the study, i.e. a minimum of 2 episodes. We anticipate that most young people will be self-harming at a higher frequency. Please note that this study was extensively reviewed by our funders (National Institute of Health Research) who reviewed our methods, eligibility criteria and outcomes and are assured that our methods will answer our question.

Reviewer: 2

Dr. Alberto Forte, Sapienza University of Rome Comments to the Author:

I am thankful for the opportunity to review the present study protocol. I find that the present study is very important and up to date, as there is a lack of evidence-based mobile apps to be used with the aim of preventing self-harm in adolescents. Moreover, the previous study already demonstrated the feasibility and acceptability of the Bluelce app. The methodology is very clearly described, the paper is well written and readable.

- Thank you for your positive comments

I would only recommend the authors providing some more details on the importance of the present study. I would highlight the utility of mobile devices for suicide prevention in adolescents, and also the lack of studies on the effectiveness. A recent review paper highlighted this aspect and might be useful for the introduction section. Forte A, Sarli G, Polidori L, Lester D, Pompili M. The role of new technologies to prevent suicide in adolescence: a systematic review of the literature. *Medicina*. 2021 Feb;57(2):109.

- Thank you for this suggestion. We have added the following paragraph to highlight the importance of this study and included the suggested reference.

"In terms of self-harm and suicide prevention, mhealth appears very acceptable and appealing to young people and offers a way of providing immediate support at times of crisis. However, hardly any suicide or self-harm prevention apps have been developed with a recent systematic review identifying only four [29]. Similarly, systematic reviews evaluating the efficacy of technology and mhealth interventions in preventing suicide or self-harm in adolescents [30] and university students [31] have raised similar concerns. In addition, the availability of self-harm prevention apps is extremely limited with few being available for clinical use [31]. Whilst these reviews all highlight the potential of self-help apps, further research is required to determine their efficacy in self-harm and suicide prevention with clinical populations."

I would also suggest moving the description of Bluelce in the method section rather than the introduction, as I would use the introduction to specify the importance of the present study given the lack of such investigations.

- We have removed this section from the introduction

Also, the introduction is weak in providing information and importance of the problem; self-harm is not only a risk factor for suicide attempts, but those who self-harm are at risk for later completed suicide (more recent citation: Hawton, K., Bergen, H., Cooper, J., Turnbull, P., Waters, K., Ness, J., & Kapur, N. (2015). Suicide following self-harm: Findings from the Multicentre Study of self-harm in England, 2000-2012. *Journal of Affective Disorders*, 175, 147–151.

- Thank you for this suggestion. We have added the following and included the reference you suggest: "Studies have demonstrated that adolescent non-suicidal self-harm is a strong predictor of

future suicide attempts in young adults [10]. Findings such as these highlight the importance of intervening with those who are self-harming in preventing future suicides (11)”

Nonetheless, these aspects might be further included in the discussion section, which seems to be too synthetic.

- Following this comment, we have revised the discussion. “Intervening with those who self-harm is important since self-harm during adolescence is a significant risk factor for future suicide attempts and completed episodes. The development of mHealth apps to support mental health interventions for adolescents offers a novel and accessible way of providing support at times of crisis. However, whilst technology offers many potential benefits, few apps to prevent self-arm have been developed and there is a lack of evidence regarding their efficacy, safety and acceptability”.

VERSION 2 – REVIEW

REVIEWER	Forte, Alberto Sapienza University of Rome
REVIEW RETURNED	27-Jun-2021
GENERAL COMMENTS	The authors properly provided answers to my minor comments. I do endorse the publication of the MS in its present form.